# GOAL: Gist-set Online Active Learning for Efficient Chest X-ray Image Annotation

**Chanh D. T. Nguyen**[*1,2]                 V.CHANHNG@VINBRAIN.NET, CHANH.NDT@VINUNI.EDU.VN
**Minh-Thanh Huynh**[*1]                                V.THANHHUYNH@VINBRAIN.NET
**Tran Minh Quan**[1,2]              V.QUANTRAN@VINBRAIN.NET, QUAN.TM@VINUNI.EDU.VN
**Ngoc Hoang Nguyen**[1]                                   V.HOANGNG@VINBRAIN.NET
**Mudit Jain**                                                MUDITJAI@GMAIL.COM
**Van Doan Ngo**[3]                                          V.DOANNV1@VINMEC.COM
**Tan Duc Vo**[4]                                             DUC.VT@UMP.EDU.VN
**Trung H. Bui**                                             BHTRUNG@YAHOO.COM
**Steven QH Truong**[1]                                     BRAIN01@VINBRAIN.NET
[1] *VinBrain, Vietnam*          [2] *VinUniversity, Vietnam*          [3] *VinMec Hospital, Vietnam*
[4] *University Medical Center of Ho Chi Minh City, Vietnam*

## Abstract

Deep learning in medical image analysis often requires an extensive amount of high-quality labeled data for training to achieve human-level accuracy. We propose Gist-set Online Active Learning (GOAL), a novel solution for limited high-quality labeled data in medical imaging analysis. Our approach advances the existing active learning methods in three aspects. Firstly, we improve the classification performance with fewer manual annotations by presenting a sample selection strategy called gist set selection. Secondly, unlike traditional methods focusing only on random uncertain samples of low prediction confidence, we propose a new method in which only informative uncertain samples are selected for human annotation. Thirdly, we propose an application of online learning where high-confidence samples are automatically selected, iteratively assigned, and pseudo-labels are updated. We validated our approach on two private and one public datasets. The experimental results show that, by applying GOAL, we can reduce required labeled data up to 88% while maintaining the same F1 scores compared to the models trained on full datasets.

**Keywords:** Deep Learning, Active Learning, Chest X-ray.

## 1. Introduction

Chest X-ray (CXR) is one of the most popular and important imaging examination methods for screening, diagnosing, and managing public health. However, the clinical interpretation of a CXR image requires the expertise of highly qualified radiologists whose diagnosis may include potential biases. Firstly, the geographical bias where some diseases appear more frequently in specific areas but are very rare in some others. Secondly, expertise bias where radiologists are only good at diagnosing a specific set of diseases. Thirdly, inconsistency among radiologists, especially on ambiguous edge cases, causes more noisy labeled data. The automated CXR interpretation system that assists radiologists in decision-making would, therefore, tackle these problems.

An automated CXR interpretation software, at the level of an experienced radiologist, could provide a great benefit in both consistency and efficiency of diagnosis. However,

---

\* Contributed equally

it poses a computational challenge to develop software that matches the expertise and experiences of practicing radiologists. Taking recent advantage of Artificial Intelligent (AI) and Deep Learning (DL), many systems can surpass the human-level decision in terms of accuracy in a number of computer vision tasks. However, DL, in particular supervised learning, generally requires large-scale and high-quality labeled datasets to achieve the human level of accuracy. Most importantly, these datasets are not easy to obtain in practice. This is due to the expertise required to label a large amount of data, and doctors' consensus cannot be reached easily. Thus, acquiring a high-quality and large labeled dataset is costly and time-consuming.

Several publicly available CXR datasets can be used for image classification task, for example, CheXpert (Irvin et al., 2019), which contains 224,316 chest radiographs collect from 65,240 patients in Stanford Hospital. Overall, these CXR datasets used information in doctor's notes, extracted by Natural Language Processing (NLP), as a ground-truth for training and validating any proposed Machine Learning (ML) models. However, this technique has a limitation in dealing with multi-language ambiguity and uncertainties in radiology reports. Furthermore, most of the annotations are not validated by radiologists or professional physicians to ensure the annotations' quality. Therefore, it leads to the decreasing of confidences in labels extracted from radiologist's notes. Majkowska et al. (Majkowska et al., 2020) proposed a procedure to obtain qualified labels. However, this method only produces high-quality labels but also consumes a lot of time and cost; hence, it is only well-suited for making the high-quality test and validation set.

Active Learning (AL) is a promising method to solve limited, highly qualified labeled data in the medical domain. AL mainly lies in evaluating the informativeness of data points. The main families of informativeness measurement in AL are uncertainty, Cost-Effective Active Learning (CEAL) (Wang et al., 2017), and representation, Suggestive Annotation (Yang et al., 2017).

In this work, we study the effect of the AL methods in the regime of a large and small amount of available unlabeled data. We present a novel AL method, called Gist Set Online Activate Learning (GOAL), for efficient annotations. Our approach further saves annotation costs by reducing the amount of data that needs to be additionally labeled by doctors while keeping the same performance as using full data. Our method shares a similar flow with CEAL but is different from it in two aspects. Firstly, uncertainty and representation are combined for sample selection, which we call the Gist-set Selection. Secondly, the pseudo-labels are updated using momentum after each iteration, which we call Online Active Learning. We evaluated our method based on both our private and public datasets. The private dataset consists of two findings, **68,959 positive instances** of Airspace Opacity (AO) and **12,848 positive instances** of Lung Lesion (LL) out of **131,030 annotated instances**. For the public domain, we use Pneumonia (PN) data from RSNA Pneumonia dataset[1], which contains **9,555 positive instances** out of **26,684 instances**, and Pleural Effusion (PE) from CheXpert (Irvin et al., 2019), which contains **86,477 positive instances** out of **191,027 frontal instances**.

---

1. https://www.kaggle.com/c/rsna-pneumonia-detection challenge

## 2. Method

### 2.1. Decision Boundary

A typical approach to AL (Wang et al., 2017) is to get an uncertainty using entropy measure

$$\mathcal{U}(x) = \sum_{c=0,1} -p(c|x)\log(p(c|x)) \tag{1}$$

followed by assigning label to low uncertain score instances, and randomly sample high uncertain instances for human annotation. Such approach is the same as sampling high confidence instances[2] to assign label and low confidence instances for human annotation. However, high confidence instances only yield small information gain since its feature vector lies deep above/under the decision boundary (Fig. 1). In contrast, instances whose feature vectors are in the neighborhood of the decision boundary present the most uncertain for the model to assign a specific label, thus they are the most informative for further learning.

### 2.2. Gist Data Point

Furthermore, our aim is to reduce the amount of data that needs to be annotated. In other word, we want to select only those data points that represents the global structure of the decision boundary neighborhood. However, due to the complexity of a deep CNN, the global structure of the result feature space is not well understood. Despite that, deep CNN was shown to be a good feature extractor that maps an image to an embedded high dimensional sphere (Schroff et al., 2015). Based on that, we hypothesize that the general feature space of a deep CNN can be treated as an embedded manifold inside a flat Euclidean space[3]. Using that hypothesis, we first build a local neighborhood around each data point in the neighborhood of the decision boundary using the method from (Rahmah and Sitanggang, 2016) which is quite robust, compared to global structure of the feature space. We, then, define gist points as each of them has at least some minimum amount of data points inside its neighborhood (Rahmah and Sitanggang, 2016). Finally, we sample from the set of gist points to reduce the amount of data.

### 2.3. Online Learning with Momentum

Labeled data is inherently noisy at the begining of an AL iteration, the model may not learn enough feature to generate consistent label e.g. high confidence instances in an iteration may become low confidence instances in the next one. Therefore, we adopt the approach of using a running average to stabilize the output of the model.

$$\hat{p}_t = \mu\hat{p}_{t-1} + (1-\mu)p_t \tag{2}$$

where $\hat{p}_t$ is the confident score after applying momentum modification, $p_t$ is the original confidence score of the model in iteration $t$ and $\mu$ controls how much past score affects the final score. For $\mu \in [0, 0.4]$, $\hat{p}_{t-1}$ has little effect on the final confidence score, therefore

---

2. Instances with high $p(c=1|x)$ or $p(c=0|x)$

3. The reason we assume embedding instead of immersion is because we want images with different visual to have distanced feature vector

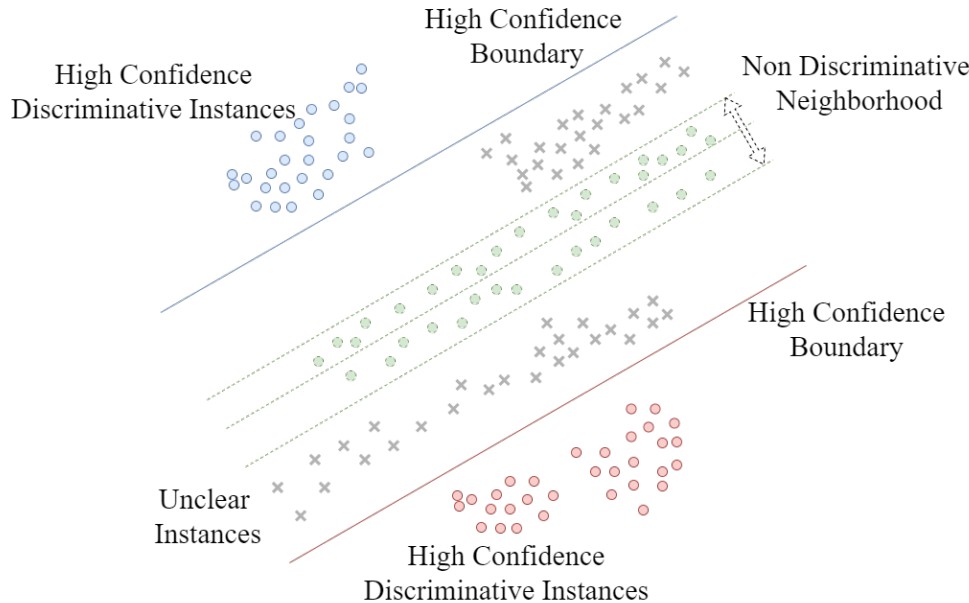

Figure 1: High confidence sample are far from decision boundary, so they're not as informative as those that are near the boundary.

instances with small fluctuation in $p_t$ will get removed from the pseudo-labeling process. Such aggressive removal is uncalled for since instances with $p_t \in [0.8, 0.9)$ but $\hat{p}_{t-1} \geq 0.9$ can be treated as high confidence data point.

In contrast to that, for $\mu \in [0.6, 1]$, instances with $p_t < 0.6$ may end up with $\hat{p}_t \geq 0.9$, which can destabilize the training process because unstable instances are being kept in the training set. Therefore we set $\mu = 0.5$ in our experiment.

## 3. Experimental Results

### 3.1. Implementation Details

For network architecture, we use ResNeXT50-32x4d (Xie et al., 2017) as our pre-trained backbone. We use SGD with Nesterov momentum (Sutskever et al., 2013) as optimizer and train the network for 8 epochs. The learning rate and scheduler is selected using the procedure from one-cycle policy (Smith and Topin, 2017) for fast convergence. Since medical data is inherently unbalanced, we follow the weighting scheme of (Cui et al., 2019) to make training more stable.

For DBSCAN feature vector, we use the output vector of the trained backbone. We follow (Rahmah and Sitanggang, 2016) to pick the radius neighborhood of each data instance. For the minimum number of data points of gist instances, we find that taking the inflection point of the previous step and divided by 10 works well (see Fig. 2).

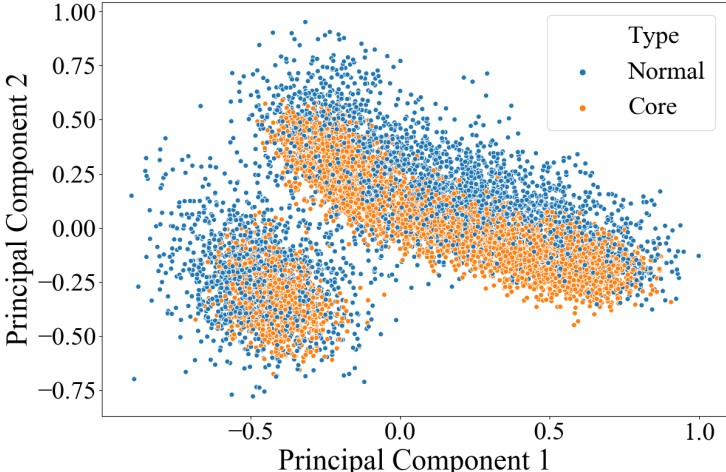

Figure 2: Lung Lesion CXR instances with $\hat{p}_t \in [0.4, 0.6]$. Blue dots represent normal data point, orange dots represent gist points which cover almost all instances.

### 3.2. Datasets

We study the AL's effectiveness in 3 cases: **1.** when the number of unlabeled data is abundant; **2.** when the number of unlabeled data is few; **3.** when NLP extracts the label from the medical report.

*For the 1st case*, we use our private datasets of AO and LL. The train set consists of $131,030$ data points with $68,959$ positive AO instances and $12,848$ positive LL instances. Two or three radiologists annotated each instance. We chose these radiologists from a pool of more than 30 radiologists with at least five years of experience. The test set consists of $4,279$ instances. Each instance is annotated by a group of five to eight radiologists from the same pool. It contains $1,789$ positive AO instances and $857$ positive LL instances.

*For the 2nd case*, we use the public RSNA Pneumonia dataset[4]. It consists of 26,684 data points, we then stratified split the data with a ratio of 9:1. The final training set consists of 24,015 data points with 5,411 positive instances, and the final test set consists of 2,669 data points with 601 positive instances.

*For the 3rd case*, we picked PE, the finding with the most positive samples, from the CheXpert dataset to study the difference between NLP labels and pseudo labels. For PE, we use the U-One (Irvin et al., 2019) approach to assign positive labels to uncertainty instances. We then test the models on the same test set as the 1st case. The test set consists of $4,279$ instances with 865 positive samples. Each instance was also annotated by a group of five to eight radiologists.

---

4. https://www.kaggle.com/c/rsna-pneumonia-detection challenge

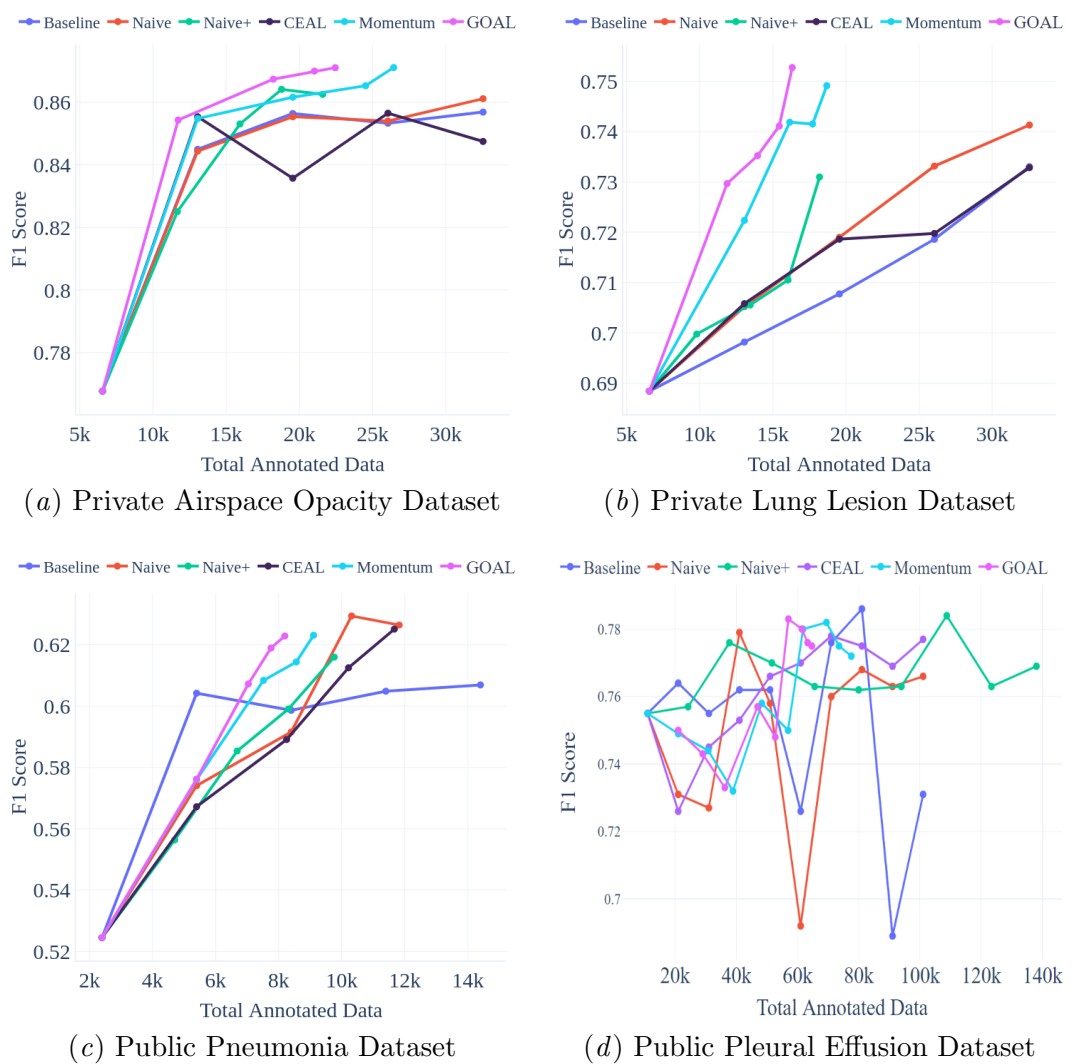

Figure 3: Performance gains of AL methods on private Airspace Opacity dataset, private Lung Lesion dataset and public Pneumonia dataset. Compare to other methods, GOAL has the most performance gain per data annotated.

### 3.3. Analysis

We retrospectively study how much data is needed for a model using AL acquired data to reach the same performance as a model using full data. We construct many AL pipelines to compare with GOAL. The details is as follows.

**Baseline**: We sample uniformly 10K instances for annotation each iterations.

Table 1: F1 score performance of Airspace Opacity (AO), Lung Lesion (LL), Pneumonia (PN), and Pleural Effusion (PE) from the CheXpert dataset. We ran 10 iterations to investigate the effect of NLP and pseudo labels

| Method | Finding | \multicolumn Active Learning Iteration | | | | | Finding | \multicolumn Active Learning Iteration | | | | | | | | | |
|---|---|---|---|---|---|---|---|---|---|---|---|---|---|---|---|---|---|
| | | 1 | 2 | 3 | 4 | 5 | | 1 | 2 | 3 | 4 | 5 | 6 | 7 | 8 | 9 | 10 |
| Full | | | | **0.871** | | | | | | | | **0.631** | | | | | |
| Baseline | | | 0.845 | 0.856 | 0.853 | 0.857 | | | 0.604 | 0.599 | 0.605 | 0.607 | - | - | - | - | - |
| CEAL | | | 0.855 | 0.836 | 0.857 | 0.848 | | | 0.567 | 0.589 | 0.613 | 0.625 | - | - | - | - | - |
| Naive | AO | 0.768 | 0.844 | 0.855 | 0.854 | 0.861 | PN | 0.525 | 0.574 | 0.592 | 0.612 | 0.626 | - | - | - | - | - |
| Naive+ | | | 0.825 | 0.845 | 0.845 | 0.845 | | | 0.556 | 0.585 | 0.616 | 0.620 | - | - | - | - | - |
| Momentum | | | **0.855** | 0.862 | 0.865 | **0.871** | | | 0.576 | 0.608 | 0.614 | 0.623 | - | - | - | - | - |
| GOAL | | | 0.854 | **0.867** | **0.870** | **0.871** | | | **0.576** | **0.607** | **0.619** | 0.623 | - | - | - | - | - |
| Full | | | | 0.743 | | | | | | | | | 0.785 | | | | |
| Baseline | | | 0.698 | 0.708 | 0.719 | 0.728 | | **0.764** | 0.755 | 0.762 | 0.762 | 0.726 | 0.776 | **0.786** | 0.689 | 0.731 | |
| CEAL | | | 0.706 | 0.719 | 0.720 | 0.737 | | 0.726 | 0.745 | 0.753 | 0.766 | **0.770** | 0.778 | 0.775 | 0.769 | **0.777** | |
| Naive | LL | 0.688 | 0.705 | 0.719 | 0.733 | 0.741 | PE | 0.731 | 0.727 | **0.779** | 0.758 | 0.692 | 0.760 | 0.768 | 0.763 | 0.766 | |
| Naive+ | | | 0.700 | 0.706 | 0.711 | 0.735 | | 0.757 | 0.776 | 0.770 | 0.763 | 0.762 | 0.763 | 0.784 | 0.763 | 0.769 | |
| Momentum | | | 0.722 | **0.742** | **0.742** | 0.749 | | 0.749 | 0.744 | 0.732 | 0.758 | 0.750 | 0.780 | 0.782 | 0.775 | 0.772 | |
| GOAL | | | **0.730** | 0.735 | 0.741 | **0.753** | | 0.750 | 0.743 | 0.733 | 0.757 | 0.748 | **0.783** | 0.780 | **0.776** | 0.775 | |

**CEAL:** We follow the author's process to assign labels to extremely high confidence instances $p_t \in [0, 0.001] \bigcup [0.999, 1]$. For example, with lower confidences in the complementary interval, 10K were sampled uniformly for annotation.

**Naive**: We expand CEAL's pseudo label interval to include more high confidence instances (i.e., $p_t \in [0, 0.1] \bigcup [0.9, 1]$).

**Naive+**: Based on Naive, we further refine the sampling interval for data annotation to the most informative interval $p_t \in [0.4, 0.5]$.

**Momentum**: We refine Naive+ method by replacing model's output probability with a running average of previous AL iterations. The running average is calculated using Eq. 2.

**GOAL**: Finally, we reduce the total amount of annotated data of the Momentum method by only selecting representative data points.

For all datasets, we sample an initial train set of $6,550$ instances from the original train set. For each learning iteration, we sample a maximum of $6,500$ instances from the remaining train pool to add to the current train set[5]. As shown in Table 1, for the AO dataset, the performance of AL methods gets better as we refine CEAL into GOAL, the final AO F1 score gain going from CEAL to GOAL is 2.3% while reducing the amount of data used by CEAL from 24.84% to 17.13% (Table 2). The GOAL method achieve best performance gain per annotated as shown in Fig. 3(a).

For LL dataset, there's a dip of 0.6% in going from Naive to Naive+ method, we hypothesize that sampling from uncertainty region for an unbalanced class would result in drastic change on old pseudo label. Therefore, when we use momentum to stabilize the pseudo label, the final F1 score take a drastic increase from 0.735 to 0.749 The final GOAL method achieves the best performance per annotated data as shown in Fig. 3(b).

On PN dataset, all AL approaches achieve the same comparable F1 score, we hypothesize this to be the result of lacking pseudo-labeled instance. Despite that, GOAL only uses 34.12% of the total data while CEAL needs to use 48.58% (Table 2) to achieve the same performance.

---

5. Each AL approach has a different way to sample instances in the big training set but all of them use the same initial set of training data

Table 2: Number of annotated data for each methods. The lowest amount of annotated data are in bold.

| Method | Finding | #Neg. | #Pos. | Total | % | Finding | #Neg. | #Pos. | Total | % |
|---|---|---|---|---|---|---|---|---|---|---|
| Full | | 68,959 | 62,071 | 131,030 | 100.00 | | 18,604 | 5,411 | 24,015 | 100.00 |
| Baseline | | 17,199 | 15,351 | 32,550 | 24.84 | | 11,005 | 3,393 | 14,398 | 59.95 |
| CEAL | | 20,135 | 12,415 | 32,550 | 24.84 | | 8,996 | 2,671 | 11,667 | 48.58 |
| Naive | AO | 18,342 | 14,208 | 32,550 | 24.84 | PN | 9,036 | 2,785 | 11,821 | 49.22 |
| Naive+ | | **11,678** | **9,912** | **21,590** | **16.48** | | 7,337 | **2,425** | 9,762 | 40.65 |
| Momentum | | 13,799 | 12,642 | 26,441 | 20.18 | | 6,579 | 2,528 | 9,107 | 37.92 |
| GOAL | | 11,844 | 10,603 | 22,447 | 17.13 | | **5,770** | 2,426 | **8,196** | **34.12** |
| Full | | 118,182 | 12,848 | 131,030 | 100.00 | | 104,550 | 86,477 | 191,027 | 100.00 |
| Baseline | | 26,946 | 5,604 | 32,550 | 24.84 | | 55,093 | 45,951 | 101,044 | 52.90 |
| CEAL | | 27,699 | 4,851 | 32,550 | 24.84 | | 76,030 | 68,301 | 144,331 | 75.56 |
| Naive | LL | 26,843 | 5,707 | 32,550 | 24.84 | PE | 55,234 | 45,810 | 101,044 | 52.90 |
| Naive+ | | 13,118 | **4,271** | 17,389 | 13.27 | | 75,441 | 62,621 | 138,062 | 72.27 |
| Momentum | | 13,466 | 5,224 | 18,690 | 14.26 | | 45,309 | 32,273 | 77,582 | 40.61 |
| GOAL | | **11,477** | 4,848 | **16,32** | **12.46** | | **36,792** | **27,912** | **64,704** | **33.87** |

We study the effect of active learning methods on CheXpert, the dataset with NLP generated annotation. Fig. 3(d) shows that CEAL, momentum, and GOAL methods perform more stable than other methods. We hypothesize this more stable performance is due to the consistency of using a small amount of extremely high confidence instances in CEAL and stable confidence instances in momentum and GOAL. Furthermore, the unstable performance of all methods comes from only using NLP generated annotation. Therefore, manual annotation is required if active learning methods are to be applied in medical domain.

## 4. Conclusions

We propose a simple yet novel active learning algorithm to support practitioners to acquire additional annotated CXR images efficiently. The results show that GOAL achieves full data training performance while using only $12.5\% \sim 34.1\%$ of the available annotated data. Because only data points representing the informative region are sampled, the amount of data needed decreases dramatically. Furthermore, the usage of momentum helps stabilize the training by potentially removing the noisy label and keeping only stable and high confidence data points. We have not evaluated the proposed method on other public datasets in the general domain because the primary purpose of developing this is for the medical industry. In future work, we intend to study in-depth how the representation of deep CNN affects data choice and further evaluate our work on other domains.

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
