# OpenReview forum: "GOAL: Gist-set Online Active Learning for Efficient Chest X-ray Image Annotation"
_MIDL.io/2021/Conference — MIDL 2021_

### Official Review · AnonReviewer4 · 2021-03-08

**Confidence:** 5
**Preliminary Rating:** 3
**Recommendation:** Poster
**Final Rating:** 3

**Summary:**

This paper proposed an active learning solution GOAL, which consists of three ideas: 1) a gist set sample selection strategy2) an informative sample selection that optimizes uncertain samples selection 3) online model updating strategy by assigning pseudo labels for high-confidence samples                        .

**Strengths:**

clear and logical writing, clear orgnize of experiments, clear figures .
                       .

**Weaknesses:**

1. Not enough literature on AL-related tasks, such as classification, segmentation. There are many tasks that utilize informativeness and representativeness properties to select data, these previous methods' introduction must cover at least one para in the introduction section.
2. The methods design still needs more analysis and insights, the experiment should be complementary analysis.

**Deanonymize Review:**

no

**Detailed Comments:**

1. the implementation of the CEAL method in terms of unlabeled sample selection is different from the original paper, as the CEAL  concern three aspects of uncertainty. From the experimental description of the method, I found that you only rely on the prediction probability for an unlabeled sample selection of CEAL, can you please explain why you do not consider the other two metrics when reproducing?
2. using DBSCAN for boundary sample selection refinement is quite interesting. Please add more details of utilizing this clustering method, because it has a lot of hyperparameters, such as the distance, the density numbers. The different datasets would have different parameter settings, I guess. Besides, I think it would be better when doing the DBSCAN  in high confidence pseudo labels, cause it would raise the diversity of the pseudo label, the same reason as the uncertain boundary label. It's just a suggestion for your future work.
3. I am confused about why the Momentum would work, from my perspective of view, it would not correct the probability when the model is weak at the beginning. Think about one situation that the model predicted 0.3 at time $t-1$  and 0.7 at time $t$, another situation that the model predicted 0.7 at time $t-1$  and 0.3 at time $t$, the Momentum  is not work in this situation, we still don't if $t-1$ is the right prediction or $t$. Only one situation is that could make Momentum work is that the right $0.7$ is more than the wrong $0.3$. So the network would boost to be better. So, can you show the distribution of each probability level in each iteration? Also, can you please add an ablation study on the $\mu$ choosing?
4.  I think the experiment design is quite good but lack of analysis, such as the performance increasing speed of the AL method, why the performance of some method would increase fast at the beginning, why in fig3(c), all the AL method is not working(less than the performance of random sampling).

**Final Rating Justification:**

The authors gave no response so I keep making the original decision.

**Justification Of The Preliminary Rating:**

The AL strategy of the author includes two novelty: 1) using DBSCAN for boundary sample selection refinement 2) using Momentum for pseudo labeling. The first is quite interesting, the second still needs clear analysis and quantity evidence.

**Paper Type:**

validation/application paper

**Questions To Address In The Rebuttal:**

1. Given evidence about comments 3.
2.  Give more analysis to experiments.

**Special Issue:**

no

---

### Official Review · AnonReviewer1 · 2021-03-09

**Confidence:** 3
**Preliminary Rating:** 3
**Recommendation:** Oral, Poster

**Summary:**

They proposed Gist-set Online Active Learning (GOAL), a novel solution for limited high-quality labeled data in medical imaging analysis.
They also proposed an application of online learning where high-condence samples are automatically selected, iteratively assigned, and pseudo-labels are updated.
They could reduce required labeled data up to 88% while maintaining the same F1 scores compared to the models trained on full datasets.

**Strengths:**

They proposed active learning algorithm reducing labeling burden, which is one of major issue that is should be resolved in all kinds of domain.

The algorithm that authors proposed could be used in any domain not only for the mdedical image such as X-ray.

**Weaknesses:**

Various public datasets in the general domain should be used as another extra validation set.

Authors used DBSCAN  algorithm to extract output vector of the trained backbone, which could led to better performance in terms of quantitative measurements. However, it shows lack of explaination for better performance when applying this algorithm to medical domain. For examle, the local neighborhood of the gist point can be interpreted with the needs of medical domain.

Lack of the repeated test.

**Deanonymize Review:**

no

**Justification Of The Preliminary Rating:**

They proposed an effective active learning algorithm to reduce the intensive labeling burden that should be resolved in the newest trends of deep learning in the field of medicine.

This concepts, Gist-set Online ACtive Learning might be adjusted or very useful in other domain.

**Paper Type:**

both

**Questions To Address In The Rebuttal:**

Various public datasets in the general domain should be used as another extra validation set.
Have you ever tried to adjust this algorithm to any public dataset?

Authors used DBSCAN  algorithm to extract output vector of the trained backbone, which could led to better performance in terms of quantitative measurements. However, it shows lack of explaination for better performance when applying this algorithm to medical domain. For examle, the local neighborhood of the gist point can be interpreted with the needs of medical domain.
Can you add more results showing better performance in the point of clinical routine practice?

Lack of the repeated test. P-value for showing significant high performance should be tested with the repeated test.


**Special Issue:**

no

---

### Official Review · AnonReviewer2 · 2021-03-09

**Confidence:** 3
**Preliminary Rating:** 2
**Final Rating:** 2

**Summary:**

The authors propose a solution to the high cost of manual annotations and labelling of medical imaging. The proposed method works by selecting only informatively uncertain samples for manual annotation. Labels with high certainty are automatically labelled.

The method builds on the idea that samples close to the decision boundary will be more informative when it comes to labelling. And that acquiring the "ground truth" for these labels is more valuable than for samples far from the decision boundary.

The authors test this on a large chest X-ray dataset and show promising results over alternative methods.

**Strengths:**

The paper is mostly well written and clearly sets out to explain and propose a solution to the problem selecting valuable samples for annotations.

This is an important problem in practice and improving the performance of these models can lead to significant time and cost savings for medical professionals and institutions.

**Weaknesses:**

The results, as they are presented, do not give me significant confidence on the relative performance of this method. They say 10 iterations were run but it is not clear what metric was used in the table of results. Is it the mean? median? best run? The variance of these runs is also missing. I am not sure if three significant figures can be used here without seeing the variance. Similar applies to figure 3, the having error bars on the lines would be much more convincing.

Although the paper is mostly well written, there is high inconsistency between quality. For example, section 3.2 is significantly less polished than the rest of the paper.

**Deanonymize Review:**

no

**Detailed Comments:**

- In the introduction there is an "Anna et al." in the text
- The writing in section 3.2 needs improvement. There are spelling mistakes (e.g. the start of sentences should start with capitals) and the numbering of items is difficult to read.
- In section 3.2 it says the instances were annotated by 2-3 radiologists. Was this 2-3 radiologists per image or did 2-3 radiologists split the set between themselves (1 radiologist per image, 2-3 total).
- What is a "stratifical split" (section 3.2)?
- "COAL" in figure 3 is probably meant to be "GOAL"

**Final Rating Justification:**

I thank the authors for responding to my review.

Unfortunately, the authors have not addressed all of my concerns. One of my critical questions was not answered (it is unclear what the number in table 1 are: mean? median? best run?)

Furthermore the response created more confusion. In the paper they report using SGD and running 10 iterations, yet in the response they authors say: "After experimenting and tuning the method, we re-run all experiments from scratch and report the result. Therefore, we did not have the variance for the final experiments." Running 10 iterations implies that the trainable parameters were randomly initialised and batches were shuffled during SGD. This should result in a variance between runs and this variance should be reported. If the same random seeds were used for all runs, there needs to be a clear motivation for that.

For these two critical reasons, I will not improve my rating.

**Justification Of The Preliminary Rating:**

The paper proposes a solution to the annotation cost problem. The problem and proposed solution are explained clearly but the paper suffers from unconvincing presentation of the results. This can be improved by clarifying some of things listed above.

**Paper Type:**

methodological development

**Questions To Address In The Rebuttal:**

1. What are the numbers reported in the table? Mean? Best runs?
2. What is the variance/error of the runs?
3. Why do you think CEAL performs worse than baseline in some of your experiments? Could it be something about your dataset/setup?
4. What are some of the limitations of your work? What assumptions do you make on the distribution(s) of the data?

**Special Issue:**

no

---

### Official Review · AnonReviewer3 · 2021-03-09

**Confidence:** 3
**Preliminary Rating:** 3
**Final Rating:** 3

**Summary:**

This paper addressed the active learning problem in chest x-ray images. The authors studied thoroughly about the present state-of-the-art method CEAL and modified the network for better performance. In their modification, they used the momentum to select the data point for each AL iteration and considered uncertainty as the method to select samples in the first place. The experiments and ablation study proves their network's performance on both public and private dataset.

**Strengths:**

1. The idea to consider uncertainty is interesting and it helps to select the most meaningful points for updating the model.
2. The experiments can be recognized as solid and the ablation study can prove the effectiveness of their modifications.

**Weaknesses:**

1. The description of the methods could be more clarified. For example, the definitions of P_t and \hat{P_t} are not clear in section 2.3.
2. The improvement of the method seems to be mainly the way of selecting data points.

**Deanonymize Review:**

no

**Final Rating Justification:**

Thanks for your update. After reading the discussion I would like to keep my voting.

**Justification Of The Preliminary Rating:**

The authors showed their insights into the active learning problem and the experiments are solid to show that their modifications upon the baseline network are effective. I would rate weakly accept due to the performance of the modifications, but the level of novelty could be a challenge.

**Paper Type:**

both

**Special Issue:**

no

---

### Meta-Review · Area_Chairs · 2021-03-25

**Recommendation:** Accept (Poster)

**Metareview:**

This paper proposes an online active learning for chest x-ray image annotation. It claims three novelties: 1) it improves the classification performance with fewer manual annotations than current approaches; 2) it selects informative uncertain samples; and 3) it assigns pseudo labels for high-confidence samples  in an online model update strategy. The majority of reviewers support the publication of this paper, where the main strengths are the paper writing, the importance of active learning for annotation costs savings, and generalisation of the method to other domains. The reviewers also identified a few problems, namely lack of novelty, unclear and incomplete experimental results, lack of statistical evaluation, and poor review for active learning papers.  Despite these issues, I believe the paper is valuable for publication.

**Paper Type:**

methodological development

---

### Decision · Program_Chairs · 2021-03-31

Accept